# Low-Coherence Homodyne Interferometer for Sub-Megahertz Fiber Optic Sensor Readout

**DOI:** 10.3390/s24020552

**Published:** 2024-01-16

**Authors:** Petr Volkov, Andrey Lukyanov, Alexander Goryunov, Daniil Semikov, Oleg Vyazankin

**Affiliations:** The Institute for Physics of Microstructures RAS, Academicheskaya Str. 7, Nizhny Novgorod 603087, Russia; luk@ipmras.ru (A.L.); gorav@mail.ru (A.G.); semikovda@ipmras.ru (D.S.); o.sados@yandex.ru (O.V.)

**Keywords:** fiber optic sensor, low-coherence interferometry, homodyne demodulation

## Abstract

This study proposes a method for interferometric fiber optic sensor readouts. The method utilizes the advantages of the active homodyne demodulation technique and low-coherence interferometry. The usage of the tandem low-coherence interferometer enables modulating the reference interferometer without any changes to the sensor. This achieves high sensitivity, high stability, and a wide frequency band. A sensitivity of up to 0.1 nm (RMS) in the frequency range of 5 kHz is demonstrated by detecting acoustic signals with a fiber Michelson interferometer as a sensor.

## 1. Introduction

Currently, fiber optic sensors (FOSs) have many and varied applications in science and technology. There are many variants of FOSs for various physical quantities. The most popular are the optical fiber Bragg grating (FBG) and various types of interferometers. The FBG is widely used due to its simple manufacturing and encoding of the measured parameter [1,2]. However, direct spectral reading is slow and does not allow the use of FBG for detecting fast processes, such as acoustic signals. Some solutions for a high-frequency readout from FBG [3,4] have problems with the sensitivity and linearity of the response. The FBG also has a high sensitivity to temperature and deformation [5]. This greatly complicates the use of FBG for high-frequency process detection since deformation and temperature lead to a large shift in the entire spectral curve of the FBG and complicate signal processing. So, for fast process detection, interferometric sensors are usually used [6,7,8,9]. The main configurations of the interferometric sensors are Michelson interferometric sensors [10], Mach–Zehnder interferometric sensors [11], Sagnac interferometric sensors [12], and Fabry–Perot interferometric sensors [13]. Generally interferometric FOS are used for the detection of various physical quantities, such as temperature, pressure, etc. [14,15,16,17]. The working principle of such sensors is based on the change in the optical path difference of the optical waves in the FOS under an external influence, for example, the temperature dependence of the index of refraction [18,19], the membrane shift for pressure and vibrations [20], and so on.

A very important part of fiber optic sensing systems is the system for signal readout. There are various designs of the readout schemes for interferometric sensors. One very popular method is based on the broadband source and a spectrometer, in which a change in the sensor head length leads to a shift in the maxima and minima in the spectrum of reflected light [9,21]. The main drawback of such a scheme is the rather low readout rate of the spectrometer: 1–10 kHz. The other method is the use of wavelength scanning with a monochromatic laser [22]. This variant is very similar to the scheme with a spectrometer in terms of signal processing. For some variants of the interferometric sensors with optical path difference of about several mm, which is a typical value for temperature or deformation sensors, a small wavelength scanning range (about 1–2 nm) is needed. It allows the use of the DFB laser diodes, where the wavelength can be modulated by injection current. It makes the system very compact and inexpensive. But the typical measurement rate for such a scheme is about the same as for spectrometers.

The alternative for a spectral readout is laser interferometry, which uses a coherent monochromatic source without wavelength scanning. The variation in the sensor length leads to varying the phase shift between the light waves and the variation in the light intensity. The main drawback of this scheme is the sensor optical length drift due to temperature change or mechanical deformation. This leads to random fluctuation in the system sensitivity and makes the registration of small fluctuations nearly impossible. Much attention has been paid to the homodyne phase demodulation technique [23,24,25,26,27,28,29,30,31,32,33], due to its wide frequency band, large dynamic range, and immunity to phase noise of the laser source. The main idea of the active homodyne demodulation technique is to form a synthetic carrier due to the modulation of the optical path difference in a sensor interferometer [34], which can eliminate its drifts. However, it requires modulating the sensor directly, which significantly complicates the scheme and multiplexing.

Low-coherence interferometry is another detection technique used for fiber optic sensor readout. The main advantage of the system is its insensitivity to changes in the parameters of the transmission system [35]. It is usually used with fiber optic Fabry–Pérot interferometers and results in excellent measurement sensitivity [36,37]. Thus, a fiber-based system where the light is brought to the sensor through a single-mode fiber enables the isolation of the main measurement part from the sensors over a distance of hundreds of meters. Such systems can thus be easily adapted to industrial applications and harsh environments because the electronics can be kept away from the environment at any suitable point [38].

The paper proposes a new original method of active homodyne demodulation based on a tandem low-coherence interferometer.

## 2. Methods

The proposed method is based on low-coherence interferometry [39,40], as shown in Figure 1a.

Two interferometers with the differences in optical path length $\Delta_{1}$ and $\Delta_{2}$ are illuminated sequentially by a broadband light source with a short coherence length $L_{coh}$. The first (reference) interferometer enables the controllable adjustment of the difference in optical length. The second interferometer plays the role of a sensor. A change in the delay $\Delta_{1}$ in the first interferometer gives rise to an interference signal at the system’s output, which has two peaks. (Figure 1b).

The intensity at the output of the scheme can be described as [41]:(1)$$I\left(\Delta_{1},\Delta_{2}\right)=\frac{I_{0}}{4}\left(1+\gamma \left(\Delta_{1}\right)+\gamma \left(\Delta_{2}\right)+\frac{1}{2}\gamma \left(\Delta_{1}+\Delta_{2}\right)+\frac{1}{2}\gamma \left(\Delta_{1}-\Delta_{2}\right)\right),$$
where $\Delta_{1,2}$ are the optical path differences in the reference and sensor interferometers, $I_{0}$ is the light intensity from the SLD, and γ(Δ) is the light source autocorrelation function. The first term in (Equation 1) denotes the constant level of the light intensity, the second term is a “zero” signal (i.e., in the vicinity of $\Delta_{1}=0$), the third term is a signal from the sensor (near $\Delta_{1}=\Delta_{2}$). Therefore, by determining the position of signal II during scanning of $\Delta_{1}$, it is possible to carry out remote measurements of a sample optical thickness. More details on the TLCI operation are available in [18,40,42].

If the power spectral density of the light source has a Gaussian form,
(2)$$G\left(\nu \right)=\frac{1}{\sqrt{\pi {\left(\Delta \nu \right)}^{2}}}\exp\left(-\frac{{\left(\nu -\nu_{0}\right)}^{2}}{{\left(\Delta \nu \right)}^{2}}\right),$$
which is quite well performed for the superluminescent diodes, the autocorrelation function will have the form
(3)$$\gamma \left(\Delta \right)=\exp\left(-\frac{\Delta^{2}}{{L_{coh}}^{2}}\right)\cos k\Delta ,$$
where $L_{coh}=\frac{c}{\pi \Delta \nu }$ is the light source coherence length.

When the difference between the sensor length *d* and the reference interferometer path difference $\Delta l=l_{1}-l_{2}$ is less than the coherence length (we call it a match condition),
(4)$$|\Delta l-d|<L_{coh},$$
then (Equation 1) will have the form
(5)$$I\left(\Delta_{1},\Delta_{2}\right)=\frac{I_{0}}{4}\left(1+\frac{1}{2}\gamma \left(\Delta_{1}-\Delta_{2}\right)\right).$$

More generally, (Equation 5) can be written as
(6)I(Δl,d)=A+Bγ(Δl−d),
where $\Delta_{1}=\Delta l$, $\Delta_{2}=d$, A,B are some constants that depend on the reflection coefficients of the sensor and losses.

If we modulate the optical path difference in the reference interferometer, then we can write
(7)$$\Delta l\left(t\right)=\Delta l_{0}+\tilde{\Delta l}\cos\omega_{0}t,$$
where $\tilde{\Delta l}$ is the amplitude of the reference interferometer path difference and $\omega_{0}$ is the reference interferometer modulation frequency.

At the same time, the dependence of the optical delay in the sensor interferometer d(t) with some arbitrary external influence can be written as
(8)$$d\left(t\right)=d_{0}+\tilde{d}\left(t\right),$$
where $\tilde{d(t)}$ is the sensor length change due to the external influence.

Then, near the maximum of the autocorrelation function, where $\exp\left(-\frac{\Delta^{2}}{{L_{coh}}^{2}}\right)\approx 1$, expression (Equation 6) takes the form
(9)$$I\left(\Delta l,d\right)=A+B\cos[C\cos\omega_{0}t-\phi \left(t\right)],$$
where $C=\frac{2\pi }{\lambda }\tilde{\Delta l}$, $\phi \left(t\right)=\frac{2\pi }{\lambda }\tilde{d}\left(t\right)$.

The signal (Equation 9) has the form of the standard input signal for homodyne demodulation algorithms. We use a homodyne demodulation technique based on the differential cross-multiplying algorithm [43]. After detection and analog-to-digital conversion, the signal (Equation 9) is multiplied by the first and second harmonic of the reference modulation signal. As a result, after passing through the low-pass filters, the following signals are generated at the output of the filters:(10)$$S_{1}\left(t\right)=-BJ_{1}\left(C\right)\sin\phi \left(t\right),$$
(11)$$S_{2}\left(t\right)=-BJ_{2}\left(C\right)\cos\phi \left(t\right),$$
where $J_{1,2}\left(C\right)$ is the first- and second-order Bessel functions of the first kind.

After differentiation of the signals (Equation 10) and (Equation 11) with cross-multiplying, the result is
(12)$$S_{mult1}\left(t\right)=B^{2}J_{1}\left(C\right)J_{2}\left(C\right)\frac{d\phi (t)}{dt}\cos^{2}\phi \left(t\right),$$
(13)$$S_{mult2}\left(t\right)=-B^{2}J_{1}\left(C\right)J_{2}\left(C\right)\frac{d\phi (t)}{dt}\sin^{2}\phi \left(t\right).$$

After subtracting (Equation 13) from (Equation 12), we have
(14)$$S_{sub}\left(t\right)=-B^{2}J_{1}\left(C\right)J_{2}\left(C\right)\frac{d\phi (t)}{dt}.$$

After the final integration, the output signal has the form
(15)$$S_{out}\left(t\right)=B^{2}J_{1}\left(C\right)J_{2}\left(C\right)\phi \left(t\right).$$

It can be observed that the term *B* in (Equation 15) in accordance with (Equation 9) is proportional to the intensity of the light, so it can be affected by the light intensity disturbance. However, in the case where the frequency of the signal $\tilde{d}\left(t\right)$ is much less than the frequency of the modulation $\omega_{0}$, the term *B* can be simply extracted from the input signal with any envelope extraction algorithm, the final signal can be normalized, and thus the intensity disturbance influence will be neglected.

The difference of the Δl−d from 0 will change the output signal (Equation 15) to:(16)$$S_{out}\left(t,\Delta l,d\right)=B^{2}J_{1}\left(C\right)J_{2}\left(C\right)\phi \left(t\right)e^{-{\left(\frac{\Delta l-d}{L_{coh}}\right)}^{2}}.$$

An additional factor determined by the low-coherence source causes noticeable distortions of the algorithm when $\left|\Delta l-d\right|\approx L_{coh}$. But in the region $\left|\Delta l-d\right|\ll L_{coh}$, its influence is negligible. For typical superluminescent diodes, the coherence length is about 20–30 um, so the proposed algorithm is well suited for the signals with amplitudes less than 2–3 um.

A very important point for absolute measurements is the calibration of the system. As can be seen from (Equation 15), the output signal is proportional to the ϕ(t) or $S_{out}\left(t\right)=\beta \phi \left(t\right)$. However, for the real system, the proportional coefficient $\beta =B^{2}J_{1}\left(C\right)J_{2}\left(C\right)$ has to be measured directly in the experiment. Note that (Equation 9) was written for the case $|\Delta l_{0}-d_{0}|\approx 0$. The full variant will have the form
(17)$$I\left(\Delta l,d\right)=A+B\cos[C\cos\omega_{0}t-\phi \left(t\right)+\left(\Delta l_{0}-d_{0}\right)],$$

As can be seen from (Equation 17), if we modulate $\Delta l_{0}$ with known amplitude, then we have the calibration signal to measure the coefficient β. This can easily be accomplished because the main part of the system is the reference interferometer, with the possibility of the optical path differences being controlled and changed. This procedure must be performed immediately before the measurements.

Thus, we can use the standard active homodyne demodulation technique, but the modulation is not made in the sensor as usual but in the reference interferometer. This is the main advantage of the proposed system, which enables using it with any passive interferometric sensor.

To evaluate the main parameters of the proposed method, such as the amplitude and phase frequency response (AFR and PFR) and total harmonic distortion (THD), the mathematical modeling of the scheme with sinusoidal external influence was performed. The AFR was scored as
(18)$$AFR\left(\omega \right)=20\mathrm{lg}\frac{A_{out}\left(\omega \right)}{A_{in}\left(\omega \right)},$$
where $A_{in}$ is the amplitude of the input signal at the frequency ω, and $A_{out}$ is the amplitude of the output signal at the frequency ω.

The PFR was scored as
(19)$$PFR\left(\omega \right)=\arctan\frac{\langle H\left(S_{in}\right)\cdot S_{out}\rangle }{\langle S_{in}\cdot S_{out}\rangle },$$
where $S_{in}$ is an input signal, $S_{out}$ is an output signal, $H(S_{in})$ is the Hilbert transform of the input signal, and 〈...〉 is the period average. Note that in the standard definition, the PFR is an argument of the complex transfer function, but in the experiment for sinusoidal input, it can be scored with (Equation 19).

The THD was scored as
(20)$$THD=\frac{\sqrt{\sum_{n=2}^{\infty }A_{k}^{2}}}{\sqrt{\sum_{n=1}^{\infty }A_{k}^{2}}},$$
where $A_{k}$ is an amplitude of the k-th harmonic of the input frequency at the output of the system.

The parameters of the scheme were taken as follows:SLD central wavelength: $\lambda_{SLD}$ = 1310 nm;SLD spectrum width: $\Delta \lambda_{SLD}$ = 40 nm;Amplitude of the reference modulation $C=0.34\lambda_{SLD}$;Frequency of the reference modulation: $f_{ref}$ = 20 kHz;Sensor length modulation (signal to be detected): $\tilde{d}\left(t\right)=D\cos2\pi f_{s}t$Amplitude of the signal: $D=0.14\lambda_{SLD}$;Frequency of the signal: $f_{s}=\left(0.05-0.5\right)f_{ref}$.

In accordance with (Equation 9), the output signal is phase-modulated; so, in the common case, it has some number of the harmonics of the input signal over the noise. It leads to the limitation of the proposed method: it can work only when there is no overlap of the side harmonics with frequencies $f=mf_{ref}\pm nf_{s}$ from different orders *m* of the reference frequency (Figure 2). Of course, it is important only for the harmonics to exceed the noise level.

In the case of increasing the signal frequency, spectrum overlapping occurs (Figure 3).

Spectrum overlapping can be seen on the THD (Figure 4), AFR (Figure 5), and PFR (Figure 6) as abnormal disturbance in the curve behavior.

The peaks on the AFR, PFR, and THD are the result of the spectrum overlap described above. It leads to some limits for the frequency range. For example, for an amplitude of the signal $D=0.14\lambda_{SLD}$, the signal frequency should be $f_{s}<0.16f_{ref}$. But inside this range, the AFR is flat, PFR is about linear, and THD is about zero. To increase the frequency bandwidth, we should decrease the signal amplitude or increase the reference frequency.

## 3. Experiment

To prove the proposed concept, an experiment was conducted. Its scheme is shown in Figure 7.

Light from the superluminescent diode is split into two, connected by the fiber optic Michelson interferometers (MI) (Figure 8). The first interferometer Int1 plays the role of the reference interferometer (RI), and the second Int2 acts as a sensor (SI). Both interferometers had the same structure. The optical arms of the MIs were formed by two piezoceramic coils with optical fiber wound around them. The length of the fiber was about 8 m for every arm. At the end of the fiber, a Faraday mirror was used to eliminate the polarization distortion of the light. The diameter of the coil was 32 mm. It was important to use a circulator between them to eliminate back reflection. Changes in the optical path difference and the modulation of the reference interferometer were achieved by applying a voltage to the coils. The first step was to measure the sensitivity of the reference interferometer to the driving voltage. For this purpose, the SLD source was changed to the laser diode with a high coherence length. The amplitude of the applied sinusoidal voltage was tuned to have the optical length modulation for one wavelength of the light source, which could be easily determined through the interferometric signal on the oscilloscope. The sensitivity of the interferometers for voltage driving was about 1.2 um/V. Afterwards, the light source was changed back to the SLD, and the reference interferometer was isolated from the external sound. Thus, its difference in arm length was changed only by the application of the voltage in the experiment. The reference modulation was generated with the digital–analog converter on the multifunction card NI6212. It has a voltage amplitude sufficient to provide the $0.34\lambda_{SLD}$ optical path modulation amplitude, and this was the same for all the experiments. The frequency of the reference modulation was set to 20 kHz.

Firstly, the AFR, PFR, and THD were measured. For this purpose, the test sinusoidal signal was applied to the sensor interferometer. Its amplitude was set to provide the $0.1\lambda_{SLD}$ optical path modulation amplitude. The frequency of the signal was changed from 0.05 $f_{ref}$ to 0.5 $f_{ref}$.

The parameters of the experimental setup were as follows:SLD central wavelength: $\lambda_{SLD}$ = 1310 nm;SLD spectrum width: $\Delta \lambda_{SLD}$ = 40 nm;SLD optical power: 1 mW;RI and SI arms’ length difference: 7 mm;Amplitude of the reference modulation $C=0.34\;\lambda_{SLD}$;Frequency of the reference modulation: $f_{ref}$ = 20 kHz;Frequency of the test signal: $f_{s}=\left(0.05-0.5\right)\;f_{ref}$.

The experimental THD (Figure 9), AFR (Figure 10), and PFR (Figure 11) were obtained.

It can be seen that the experimental characteristics were very close to the ones of the model, which means that the proposed scheme can be used for signal demodulation.

Finally, some test experiments for detecting sounds with SI were conducted. For sound detection, the voltage in the SI was turned off. The calibration procedure was carried out before the measurements, as described earlier. First, the reference interferometer (Int1 on Figure 7) was calibrated with the laser light source to measure the voltage that corresponded to the $\lambda_{SLD}$ path difference shift. Afterward, the sinusoidal signal was additionally applied to the reference interferometer with frequency $f_{c}=0.1f_{ref}$ and a voltage amplitude corresponding to the 0.1 $\lambda_{SLD}$ path difference change amplitude. Thus, the calibration coefficient β was taken to calculate the real distance in nanometers after signal processing. After calibration, the additional modulation was turned off, and only the reference modulation of the reference interferometer was turned on, with the parameters described above.

There were three types of impact:Mechanical impact on the SI box (Figure 12);A finger snap at a distance of 10 cm from the SI (Figure 13);A male voice at a distance of 30 cm from the SI (Figure 14).

It can be seen that the proposed scheme was able to detect a strong exposure, such as mechanical impact to the SI box, and weak signals, such as a finger snap or a low voice. In Figure 15, the spectrum of the finger snap signal is provided. It can be seen that the scheme is able to detect up to 10 kHz (0.5$f_{ref}$), but such a range is only available for small signals with amplitudes lower than several nanometers. For large acoustic signals with amplitudes of tens or hundreds of nanometers, the frequency range was lower, at 0.15–0.25$f_{ref}$.

In Figure 16, the noise of the system can be seen. The noise had a normal distribution and was about 0.1 nm (RMS).

## 4. Discussion

FOSs are becoming more and more widespread, because they have a number of advantages over classical systems, such as high sensitivity, compactness, and electromagnetic induction immunity. One of the often-used designs of the FOS is an interference sensor, where an external influence changes the phase delay between interfering waves. For example, such a design is often used in the acoustic sensors. At the same time, for acoustic sensors, it is necessary to provide a wide band, high sensitivity to the sound wave, and insensitivity to the outside environment’s influences, such as temperature, deformation, and others.

The paper proposes an original method for readout data from interferometric FOS with a homodyne tandem low-coherence interferometry. The homodyne demodulation algorithm is well known and quite often used due to its advantages, such as a wide frequency band and immunity to sensor length drift. At the same time, it requires additional modulation inside the sensor element, which greatly complicates the construction and makes it impossible to make a universal reader for interferometric FOS. The proposed scheme, which utilizes the advantages of tandem low-coherence interferometry, allows moving the modulation from the sensor element to the reference interferometer, which can be divided from each other over a long distance by an optical fiber. As a result, we obtain the universal readout scheme, which uses the homodyne demodulation algorithm but, contrary to the standard coherent scheme, can be connected to an arbitrary interferometric FOS.

The experimental setup was based on a fiber optic MI. Its construction allows precisely the controlled adjustment and modulation of the optical path difference in the interferometer’s arms by applying voltage to the optical fiber attached to the piezoceramic coil. A sensitivity ∼0.1 nm (RMS) in the 0.1–5 kHz frequency range was demonstrated. The noise level (3σ) was about ±0.3 nm. The setup demonstrated flat AFR, linear PFR, and near-zero THD over the entire operating frequency range, which is extremely important for sound-detection tasks. The possibility of detecting a large disturbance, with an amplitude of about 1 um, and weak ones, with amplitudes of several nm, was demonstrated. Such a high dynamic range with the linearity of the PFR and the flatness of the AFR allowed recording a human voice with a fiber optic MI as a sensor with good quality.

The typical amplitudes of the acoustic emission signal range from about one nanometer to tens of nanometers [44]. However, they can be up to 1–2 um for a strong external impact [39]. The maximum suitable signal amplitude for the proposed algorithm is determined using the coherence length of the light source (which should be much less than the coherence length) and is about 2-3 um for the superluminescent diodes used. Thus, the proposed scheme has a dynamic range that is very suitable for acoustic emission, which is the most popular ultrasound diagnostic technique.

It must be noted that, for many ultrasound techniques such as acoustic emission diagnostics, frequencies up to several hundred kilohertz are used. The proposed method could be extended to frequencies much higher than those demonstrated by increasing the reference frequency. As can be seen from the results obtained for a dynamic range of about 0.15$\lambda_{SLD}$, i.e., about 200 nm, the frequency range is about 0.15 $f_{ref}$. Thus, for a 150 kHz frequency range, the $f_{ref}$ should be about 1 MHz, which can be generated and processed using standard analog–digital converters and digital signal processors. Increasing the reference frequency higher than 1 MHz will increase the complexity of the registration system and may not be rational.

The use of the broadband source will lead to some limitations for the construction of the acoustic sensor. It should have a flat spectral response (i.e., one without short peaks or valleys) throughout the spectral range of the source. This is usually achieved adequately for most variants of the interferometric sensors except for fiber Bragg gratings. The second important point is the dispersion of the sensors, i.e., the variation in the optical length for different wavelengths of the light source. The linear dispersion does not affect the results because it shifts the coherence peaks as a whole in accordance with the group dispersion mechanism. Higher orders of the refractive index dependence will lead to frequency chirp in the interference pattern and can be a source of error for the proposed algorithm. However, for most of the materials used in acoustic optical sensors, this effect is very low and can be neglected.

The presented results can be important and useful for the development of fiber optic monitoring systems. The proposed system can be used for detecting both slow processes such as temperature drift or atmospheric pressure and rather fast ones, such as sound waves and vibrations. At the same time, the system enables measuring the absolute value of the optical path difference in the nanometer range, which can be used in meteorological problems.

## Figures and Tables

**Figure 1 sensors-24-00552-f001:**
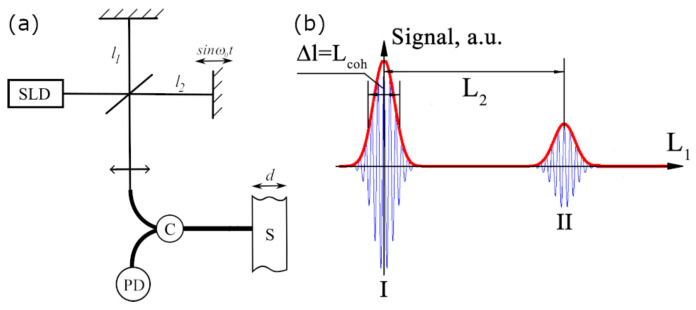
Tandem low-coherence interferometry: (**a**) TLCI scheme. (**b**) Form of a signal in the TLCI system: SLD—superluminescent diode; C—fiber optic circulator; S—sensor; PD—photodetector; d—sensor length, $l_{1,2}$—interferometer arm length; $L_{1,2}$—optical delays in the reference interferometer and sensor, respectively; $L_{coh}$—coherence length of the source.

**Figure 2 sensors-24-00552-f002:**
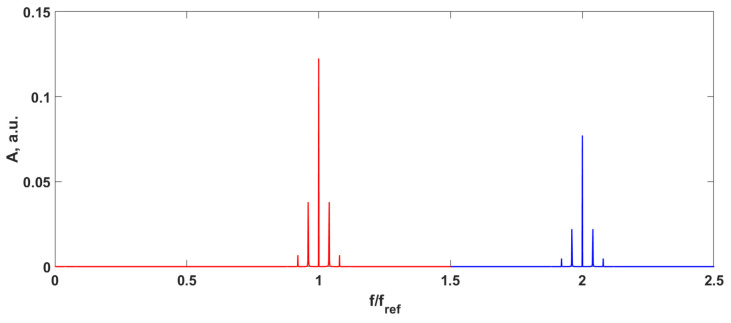
Output signal spectrum with sinusoidal input signal $f_{s}=0.1f_{ref}$. Red—the signal spectrum over the first harmonic of the reference frequency, blue—the signal spectrum over the second harmonic of the reference frequency.

**Figure 3 sensors-24-00552-f003:**
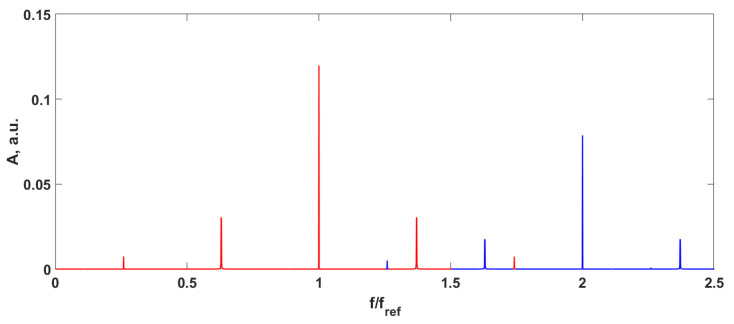
Output signal spectrum with sinusoidal signal $f_{s}=0.4f_{ref}$. Red—the signal spectrum over the first harmonic of the reference frequency, blue—the signal spectrum over the second harmonic of the reference frequency.

**Figure 4 sensors-24-00552-f004:**
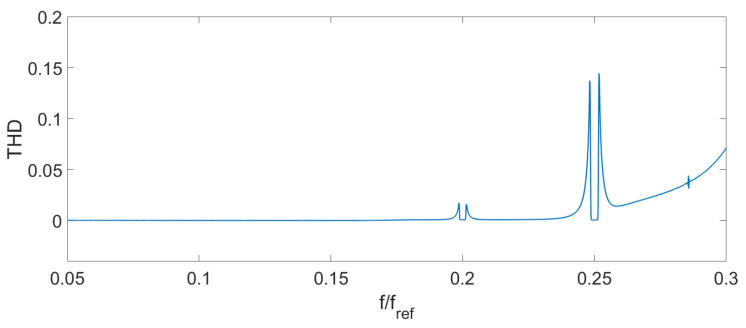
Theoretical THD $D=0.14\lambda_{SLD}$.

**Figure 5 sensors-24-00552-f005:**
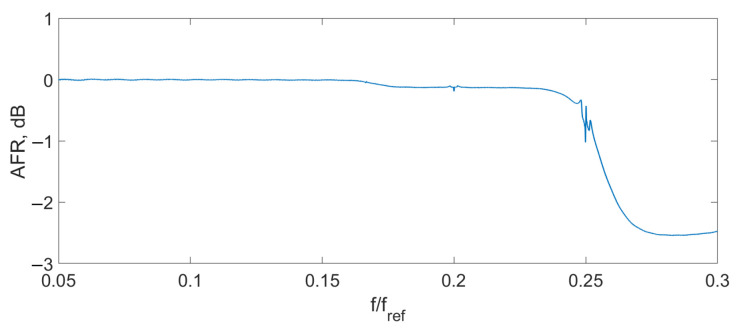
Theoretical AFR $D=0.14\lambda_{SLD}$.

**Figure 6 sensors-24-00552-f006:**
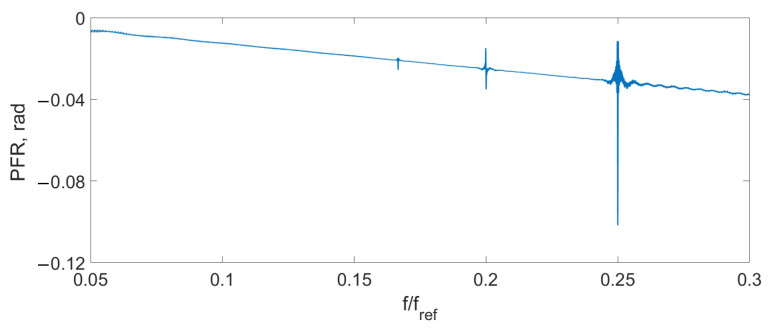
Theoretical PFR $D=0.14\lambda_{SLD}$.

**Figure 7 sensors-24-00552-f007:**
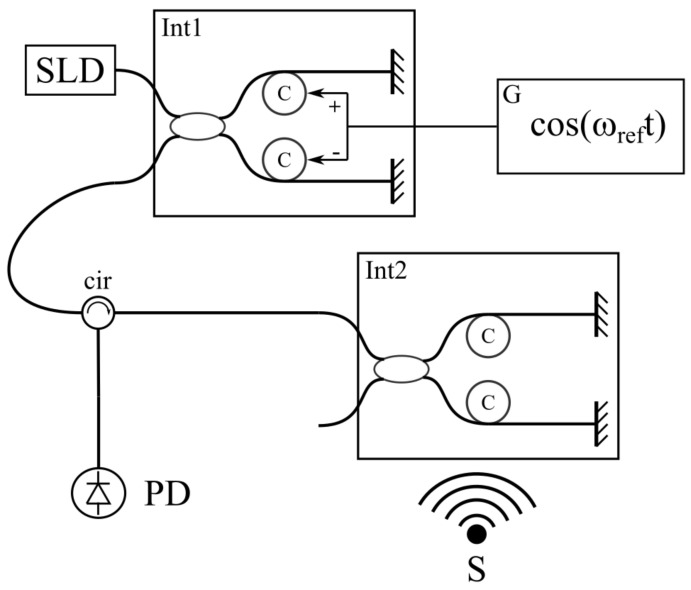
Experimental scheme. SLD—superluminescent diode, Int1—reference interferometer, Int2—sensor interferometer, C—ceramic coil with fiber optic, cir—optical circulator, S—sound source, PD—photodiode.

**Figure 8 sensors-24-00552-f008:**
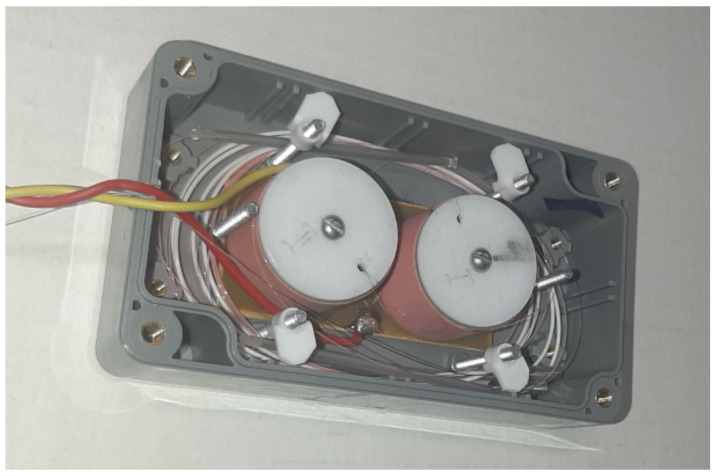
Photo of the fiber optic Michelson interferometer.

**Figure 9 sensors-24-00552-f009:**
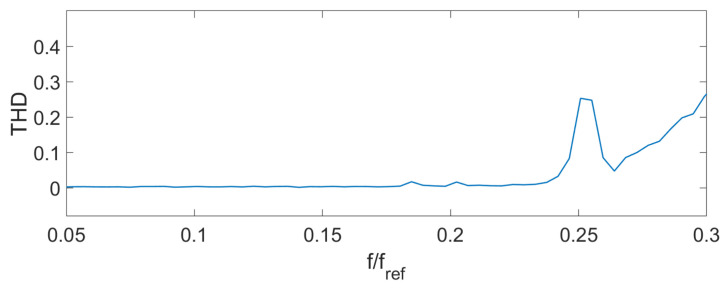
Experimental THD $D=0.1\lambda_{SLD}$.

**Figure 10 sensors-24-00552-f010:**
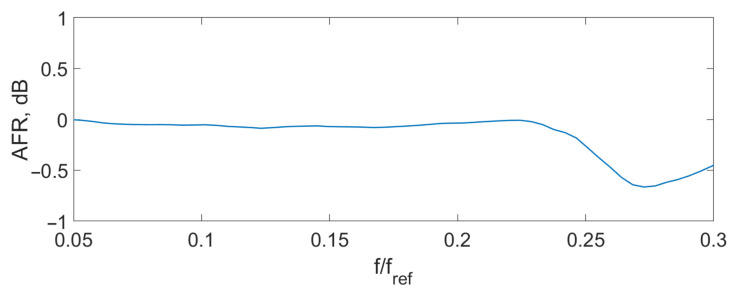
Experimental AFR $D=0.1\lambda_{SLD}$.

**Figure 11 sensors-24-00552-f011:**
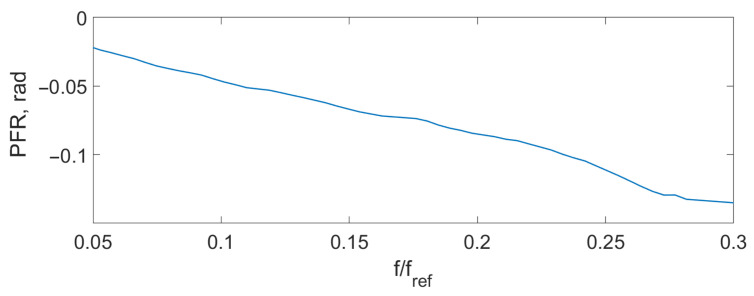
Experimental PFR $D=0.1\lambda_{SLD}$.

**Figure 12 sensors-24-00552-f012:**
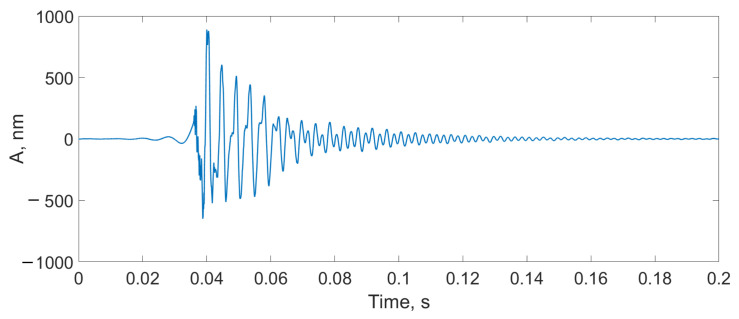
Detection of the mechanical impact on the SI box using the proposed scheme.

**Figure 13 sensors-24-00552-f013:**
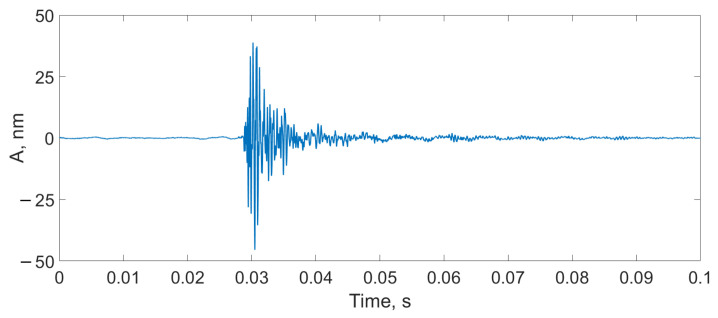
Detection of the finger snap with the SI box using the proposed scheme.

**Figure 14 sensors-24-00552-f014:**
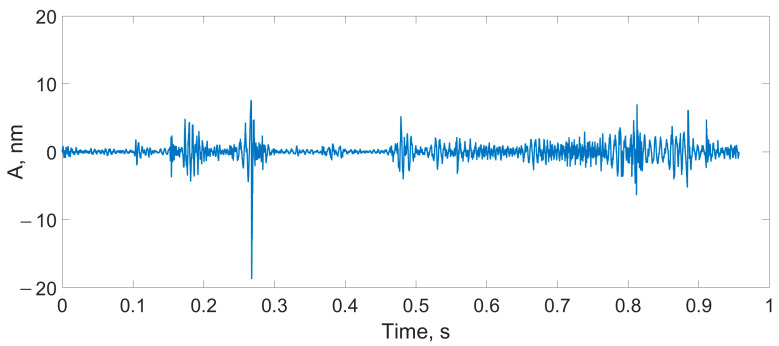
Detection of the male voice with the SI box using the proposed scheme.

**Figure 15 sensors-24-00552-f015:**
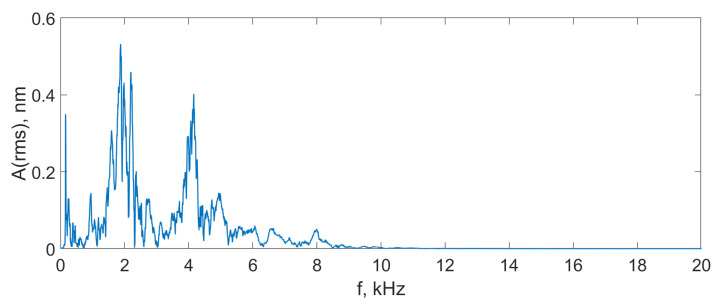
Spectrum of the finger snap signal detected using the proposed scheme.

**Figure 16 sensors-24-00552-f016:**
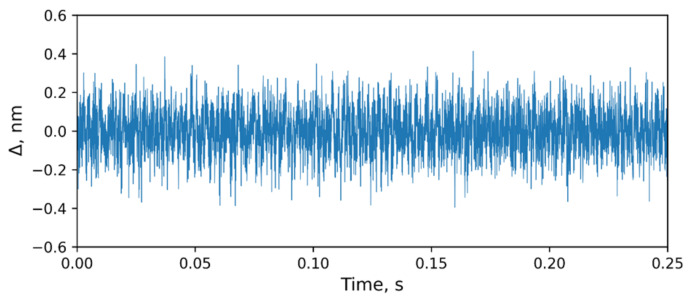
The noise of the system.

## Data Availability

All evaluated data are presented in this paper in graphical form. The raw measured data of this study are available on request from the corresponding author.

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
