# Peer review of "Low-Coherence Homodyne Interferometer for Sub-Megahertz Fiber Optic Sensor Readout"

_sensors, 2024, doi:10.3390/s24020552_

Round 1
Reviewer 1 Report
Comments and Suggestions for Authors
In this work, the authors proposed a novel interferometric fiber optic sensor readout method, which is using the homodyne tandem low-coherence interferometry. This universal readout method increases the usability of interferometric sensors. It can be accepted after the following issues are addressed:
1. Parentheses problem in writing format of equations 1 and 5.
2. Should the authors consider that the demodulation algorithm of equation 15 is affected by light intensity disturbance?
3. Is the proposed readout method applicable to larger amplitude of signal?
4. The last digit of the horizontal coordinate in Figures 2 and 3 should be regularized.
5. The accuracy of the signal demodulation in Figures 12, 13 and 14 are not verified in this paper. The numbers and labels on the axes of Figures 12, 13 and 14 are small in size.
6. The title of Figure 15 is 0.1-1kHz, but in the paper it's all about 0.1-10kHz. And Figure 15 is a time-domain plot that does not demonstrate the variation in frequency.
Reviewer 2 Report
Comments and Suggestions for Authors
This manuscript proposed a distributed active homodyne method for optical interferometric acoustic sensors and tested the idea with a fiber-based system to verify the feasibility of the proposed method. The idea of modulation at higher frequency and detect phase changes at lower frequencies is used, e.g., laser frequency locking to a reference. Of course, application to the acoustic sensors seems new to me. It might be interesting to some communities. But the paper as its present form is not suitable for publication.
Here are some major concerns:
1. The references cite heavily on ultrasound sensors, but the current manuscript only demonstrated ~10 kHz, what is the intended frequency (ies) for applications? How far in frequency range can this method be extended to?
2. The sensitivity is measured in optical path change, in fraction of a nanometer range, how does this convert to sound intensity of pressure for a typical interferometric sensor?
3. The light source is a broadband source, what are limitations for optical acoustic sensors that can used this method?
4. Since this is a distributed system, and the light source is broadband, how much does dispersion affect the method?
5. Some details of the experiment is not clear. E.g., when carrying out the test experiment, is the reference interferometer isolated from the test sound?
6. Although the range of signal frequency is claimed to be 10 kHz (0.5 ×
f_ref), none of the figures shows that. And axes labels and numbers in many figures are not legible.
7. What is the limit of the signal amplitude?
Some minor comments:
1. line 62 missing a space between two words
2. the symbol ÷ is not universally used to denote a range, may cause confusion to some readers.
From above observations, I do not recommend publication of the manuscript in its current form.
Comments on the Quality of English LanguageEnglish is ok, some paragraphs are too short, look a little weird.
Round 2
Reviewer 2 Report
Comments and Suggestions for Authors
There are some typos in the new paragraphs highlighted in yellow. E.g.,
1. Line 73, Delta_2
2. In order not to cause confusion, maybe between (5) and (6), should clearly make the substitution of Delta_1 == Delta l and Delta_2 == d.
3. Many equation numbers do not have ().
4. Line 229 and 230 betters be in the same paragraph and is the noise Gaussian distribution?
Comments on the Quality of English Language
Please go over the manuscript and correct typos carefully. Like in Line 183, "modulation for 1 wavelength" should use "one".
Author Response
We are greatly appreciate the reviewer for his comments. Please find our answers to the comments below.
Reviewer: There are some typos in the new paragraphs highlighted in yellow. E.g.,
- Line 73, Delta_2
Our Answer: We have fixed this error.
2. In order not to cause confusion, maybe between (5) and (6), should clearly make the substitution of Delta_1 == Delta l and Delta_2 == d.
Our Answer: We have added this moment in text.
3. Many equation numbers do not have ().
Our Answer: We have fixed it.
4. Line 229 and 230 betters be in the same paragraph and is the noise Gaussian distribution?
Our Answer: We have combined these lines into one paragraph. Yes, the noise had normal distribution within the accuracy of the experiment
Please go over the manuscript and correct typos carefully. Like in Line 183, "modulation for 1 wavelength" should use "one".
Our Answer: we tried to do our best and fixed some additional typos. Thank you for carefully reading the work